# It is All Connected: A New Graph Formulation for Spatio-Temporal Forecasting

## Abstract

With an ever-increasing number of sensors in modern society, spatio-temporal time series forecasting has become a de facto tool to make informed decisions about the future. Most spatio-temporal forecasting models typically comprise distinct components that learn spatial and temporal dependencies. A common methodology employs, for example, a Graph Neural Network (GNN) to capture relations between spatial locations, while another network, such as a recurrent neural network (RNN), learns temporal correlations. Alternatively, by representing every recorded sample as its own node in a graph, rather than all measurements for a particular location as a single node, temporal and spatial information is encoded in a similar manner. In this setting, GNNs can now directly learn both temporal and spatial dependencies, jointly, while also alleviating the need for additional temporal networks. Furthermore, the framework does not require aligned measurements along the temporal dimension, meaning that it also naturally facilitates irregular time series, different sampling frequencies or missing data, without the need for data imputation. To evaluate the proposed methodology, we consider wind speed forecasting as a case study, where our proposed framework outperformed other spatio-temporal models using GNNs with either Transformer or LSTM networks as temporal update functions.

## 1 Introduction

Forecasting is the task of trying to predict what will happen in the future based on current or historical information. Physical models utilise complex governing equations about dynamical systems to infer the future, while time series forecasting typically refers to extrapolating time series into the future using statistical methods. Even though one can never be certain about what is to come, forecasting has become ubiquitous in modern society to be able to make informed decisions, ranging from forecasting expected traffic flows (Yu et al., 2018; Salinas et al., 2020), items sold by retailers (Salinas et al., 2020; Qi et al., 2019), supply and demand of energy (Smyl & Hua, 2019; Salinas et al., 2020; Li et al., 2019b) and the weather (Ghaderi et al., 2017; Sønderby et al., 2020), to name a few.

Spatio-temporal forecasting leverages the fact that time series recorded at different physical locations can be heavily correlated, meaning that forecasts can be improved by considering these jointly. As an example, if a particular road in a traffic network is congested, it might affect the traffic flow on nearby roads.

With ever-increasing pressure on depleting fossil-fuel-based energy resources, the proportion of renewable energy in the electricity mix has significantly grown in recent years. Wind energy accounts for more than 840 GWs of installed capacity worldwide (gwe, 2022). Since these resources are inherently intermittent, short-term forecasting has become increasingly important to facilitate grid planning and operation. Even though spatio-temporal forecasting using deep learning (DL) has been heavily concerned with traffic network applications, such as forecasting traffic flows, ride-hailing and -sharing, recent studies are starting to shift to wind and energy forecasting (e.g. the SIGKDD Cup22 (Zhou et al., 2022a)). In this study, we decided to focus on short-term spatio-temporal wind speed forecasting. The North Sea was used as a case study since this area is planned to undergo significant developments with regard to planned off-shore wind installations in coming years (An, 2020).

Graph Neural Networks (GNN) have been popularly employed for spatio-temporal forecasting. Typically, spatial and temporal features are modelled by distinct parts of the model, such as for example, a GNN architecture to extract spatial features, with another network, such as a Recurrent (RNN) or Convolutional Neural Network (CNN), to learn temporal correlations. Furthermore, since spatio-temporal forecasting relies on measurements from a large number of sensors in different locations, there is an increased likelihood of there being missing or corrupt measurements in the available data. Since most temporal network architectures require aligned inputs, missing values are often imputed based on simple heuristics, such as mean or last recorded values, or entire samples are discarded despite there being many sensors with available information. This might deteriorate model performance as the data distribution is shifted. The main contributions of this paper can be summarised as:

- We propose a new graph formulation which allows for both spatial and temporal correlations to be learned jointly by a GNN,

- Only a GNN is required for the proposed formulation, alleviating the need for additional temporal networks such as RNNs or CNNs. Therefore, inputs do not have to be aligned, and the framework naturally allows for missing input information, irregular time series and different sampling frequencies.

- Since the proposed graph formulation removes the need for imputation or removal of samples with missing information, it does not impose additional biases and distribution shifts on the available data.

- The proposed Spatio-Temporal Unified Graph Network (STUGN) was able to outperform a range of different baselines for the task of spatio-temporal wind speed forecasting under different amounts of missing samples in the training data, proving the competitiveness of the proposed framework.

## 2 Related Works

Some of the simplest, yet very effective, methods for time series forecasting revolve around the autoregressive integrated moving average (ARIMA) model. Kavasseri & Seetharaman (2009) proposed the *fractional-ARIMA* model for one- and two-day-ahead wind speed forecasts. The RWT-ARIMA model (Singh et al., 2019) also included wavelet transform to decompose a signal into multiple sub-series with different frequency characteristics. Slightly more complicated models, such as Support Vector Regressors (SVR) or K-nearest neighbour (KNN) algorithms have also shown good forecasting performance (Jørgensen & Shaker, 2020).

Within the realms of DL, multilayer perceptron (MLP), RNN- and CNN-based models have for a long time been the most popular methods for time series forecasting. MLPs are the simplest DL models, but have shown success both when used in isolation (Sfetsos, 2002) and as part of hybrid models (Guo et al., 2011). By changing the 2D convolution operation to a 1D causal convolution, CNNs have also achieved success in time series forecasting. To be able to learn long-term context, without the need for very deep models, Oord et al. (2016) proposed to use dilated causal convolution, where the dilation can significantly increase the receptive field. This is well suited for time series and has proved effective in various wind forecasting studies (Dong et al., 2021; Shivam et al., 2020). Gated Recurrent Units (GRU) and Long Short-Term Memory (LSTM) cells are specific types of RNNs, which are both very popular for sequence analysis and time series forecasting (Yamak et al., 2019; Salinas et al., 2020; Wang et al., 2021; Lim & Zohren, 2021). Even though the LSTM and GRU cells are better at learning long-term dependencies than vanilla RNNs, these methods still rely on encoding past information into a single memory vector, which might cause information loss for very long sequences. Attention mechanisms have therefore become an integral part of more recent models that considers long time series, as dependencies can be modelled without regard to the distance (Bahdanau et al., 2014; Lim & Zohren, 2021).

The Transformer was proposed as an architecture completely without recurrence, but instead fully reliant on the attention mechanism (Vaswani et al., 2017). Transformers have taken the DL community by storm, showing impressive results for a host of different applications (Lin et al., 2022). A fundamental limitation of the vanilla Transformer is that its memory and computational complexity scale quadratically with sequence

length. Various architectures have aimed at reducing the complexity by introducing sparse attention, either based on fixed heuristics (Beltagy et al., 2020; Li et al., 2019a), or more advanced methods to locate the most important key-query pairs (Zhou et al., 2021; Wu et al., 2021). Due to the success of Transformers for various applications, these models have also translated into the forecasting domain (Pan et al., 2022; Xu et al., 2020). A range of alterations has been proposed to better facilitate time series, such as the Autoformer (Wu et al., 2021), FEDformer (Zhou et al., 2022b) and Informer (Zhou et al., 2021). However, despite the authors of the aforementioned studies reporting relative success compared to other Transformer architectures (Zhou et al., 2021; 2022b; Wu et al., 2021; Li et al., 2019a), some work also importantly question how effective Transformers really are in time series forecasting. Zeng et al. (2022) propose a very simple one-layer linear model (LTSF-Linear) and show how it was able to outperform state-of-the-art Transformer models for the long-term forecasting benchmarks used in the respective studies (Wu et al., 2021; Zhou et al., 2022b; 2021; Li et al., 2019a). Even though the study questions how effective complex Transformer models are for time series forecasting, the focus was only on long-term forecasting, 96-720 time-steps ahead, and lacked investigation into short-term forecasting. Nevertheless, Zeng et al. (2022) correctly identifies a problem with the current conduct, that many studies do not compare complex DL models against simple linear models or persistence methods.

In many domains, there are time series available for multiple physical locations, where the aim is to produce forecasts at all locations simultaneously or to improve the forecasts at a particular location by jointly considering multiple correlated series. By assigning each time series to a particular cell in a grid-like structure, CNNs were initially popular tools for extracting spatial correlations (Xu et al., 2022; Liu et al., 2020; Wang et al., 2021). Despite the success of CNNs, a problem with such formulations is the rigid ordering of the physical locations into a grid, which might not be able to best reflect more complex topologies, such as housing locations for electricity load forecasting, road networks or location of meteorological measurement stations.

Graph Neural Networks (GNNs) can better facilitate complex and dynamic ordering of the input data by processing directly on graphs. Various studies combine GNNs that learn spatial correlations, with other networks to learn temporal dependencies, such as an MLP (Pan et al., 2022), LSTM (Ghaderi et al., 2017), CNN (Wu et al., 2019) or Transformer (Cai et al., 2020). Most studies comprise a rule-based system to construct the graph adjacency matrices, while some go one step further by also learning the graph structures (Wu et al., 2020; Shao et al., 2022). Current methods usually partition the spatio-temporal forecasting problem by having separate parts of a model learn spatial and temporal dependencies separately, such as graph convolutions and recurrent networks, respectively. The StemGNN model proposed by Cao et al. (2020), applies Graph and Discrete Fourier Transform prior to a graph convolutional block to learn spatial and temporal dependencies jointly. Zhang et al. (2022) and Horn et al. (2020) do not consider forecasting, but instead, medical classification based on multivariate time series. By representing every recorded value as an individual node in a graph, Zhang et al. (2022) learn inter- and intra-series correlations jointly, while also being able to facilitate irregularly sampled time series. This methodology seems potentially interesting for forecasting, as it naturally translates to a framework for learning spatial and temporal dependencies jointly, while also being able to facilitate missing or irregularly sampled data. Since spatio-temporal forecasting utilises recorded measurements from a large number of sensors, it is often the case that samples are missing or corrupted for some sensors. With regards to wind forecasting, Tawn et al. (2020) conducted a study to investigate the effects of missing data and proposed methods for mitigating these. The study showed that missing data can have a significant negative impact on model training, but that multiple imputation can alleviate some of these issues. Wen et al. (2023) trained a model to impute missing values at the forecasting stage, showing good performance for wind forecasting in the presence of missing data. In the GRAPE framework proposed by You et al. (2020), features are encoded in the edges of a graph which naturally allows for incomplete time series.

## 3  Preliminaries

### 3.1  Traditional Problem Formulation

In a traditional spatio-temporal setting using GNNs, we have a graph; $\mathcal{G} = (\boldsymbol{X}, \boldsymbol{W})$, with node features $\boldsymbol{X} \in \mathbb{R}^{N \times T \times d_x}$, and edge weights $\boldsymbol{W} \in \mathbb{R}^{N \times N \times T \times d_w}$, where $N$ is the number of spatial locations and $T$ the number of time-steps. $d_x$ and $d_w$ are the node and edge weight feature dimensions, respectively. All observed values at timestamp, $t$, is denoted as $\boldsymbol{X}_{:,t} \in \mathbb{R}^{N \times d_x}$, while the series for a particular node, $i$, as $\boldsymbol{X}_{i,:} \in \mathbb{R}^{T \times d_x}$. For simplicity, we also assume the input graph structures and hence edge weights to be fixed across time, which results in $\boldsymbol{W} \in \mathbb{R}^{N \times N \times d_w}$. $\boldsymbol{W}_{ij} \in \mathbb{R}^{d_w}$ are the features describing the edge sending from node $i$ to $j$, e.g. the physical distance between two measurement locations, where $\boldsymbol{W}_{ij} = 0$ indicates that there is no edge connecting $i$ and $j$. Given the observed values for $T$ previous time-steps, $\boldsymbol{X}_{:,t-T}, ..., \boldsymbol{X}_{:,t-1}$ and the graph structure described by $\boldsymbol{W}$, a spatio-temporal model $f$, with parameters $\boldsymbol{\theta}$, should predict the future $K$ time-steps at the different locations as:

$$\hat{\boldsymbol{X}}_{:,t}, ..., \hat{\boldsymbol{X}}_{:,t+K-1} = f(\boldsymbol{X}_{:,t-T}, ..., \boldsymbol{X}_{:,t-1}; \boldsymbol{W}; \boldsymbol{\theta}). \tag{1}$$

### 3.2  Traditional Spatio-Temporal Framework

One of the most common types of GNNs, are message passing neural networks (MPNN), which propagate features by exchanging information amongst adjacent nodes. The operation of a single message-passing operation can be summarised as

$$h_{i,t}^{(l)} = \phi(h_{i,t}^{(l-1)}, \bigoplus_{j \in \mathcal{N}_i} \psi(h_{i,t}^{(l-1)}, h_{j,t}^{(l-1)}, e_{j,i,t}^{(l-1)})), \tag{2}$$

where $h_{i,t}^{(l)}$ are the updated features for node $i$ after layer $l$ for timestamp, $t$. We assume the inputs to the first layer are the input node and edge weight attributes, as $h_{i,t}^{(0)} = \boldsymbol{X}_{i,t}$ and $e_{i,j,t}^{(0)} = \boldsymbol{W}_{i,j}$, where the input edge weight attributes are the same for all time-steps. $\phi$ and $\psi$ are the update and message functions, e.g. linear transforms, and $\bigoplus$ is a permutation invariant aggregator, such as mean or sum. $\mathcal{N}_i$ is the set containing the neighbourhood of node $i$, i.e. $\mathcal{N}_i = \{j \forall \boldsymbol{W}_{ji} > 0\}$. Even though the input edge weights do not change over time, i.e. $e_{i,j,t}^{(0)} = e_{i,j}^{(0)}$, we denote the edge weights with a time dimension, $t$, since $e_{i,j,t}^{(l)}$ for $l \neq 0$ is generally dependent on $t$. In our traditional framework, the connectivity between nodes, i.e. $\mathcal{N}_i$, will remain fixed, but the exact edge weight features can be dynamic over time and updated, as shown by the feed-forward network (FFN) update of edge weight features in Fig. 3. The operation in eq. (2) can be stacked into multiple layers over $l$ to form a GNN. A range of different variations exist, such as Graph Attention Networks (GAT) (Veličković et al., 2017), where an attention mechanism is introduced to weigh neighbouring nodes differently, resulting in a weighted average aggregation, $\oplus$. Even though the particular graph update operations might differ slightly, the general architecture remains similar to eq. (2), with update and message functions, $\phi$ and $\psi$. The spatio-temporal frameworks using GNNs described from here on forward can use the MPNN update in eq. (2) or any other common GNN update. For more details on different types of GNNs, we refer the interested reader to Bronstein et al. (2021) and Veličković (2023).

For the MPNN update in eq. (2), aggregations and updates are only performed along the spatial dimension, considering different time-steps independently. To model temporal dependencies, another network is typically included either prior or subsequent to the message-passing update, or the update function $\phi$ is represented by a temporal network. In any case, the temporal and spatial dependencies are primarily considered separately, with the message operation only operating along the spatial dimension. Ideally, we argue that in a unified spatio-temporal framework, spatial and temporal correlations should be considered jointly.

With regards to missing values, even though the graph update in eq. (2) can handle a variable number of nodes, e.g. physical locations, the temporal update functions typically require aligned measurements and do not allow for missing values being omitted from the inputs. Therefore, missing values will have to be filled by zero-values, some more advanced data imputation or for the entire time series containing the missing information to be removed. Discarding the input series for a particular location would also mean that the model would not produce a forecast for this location.

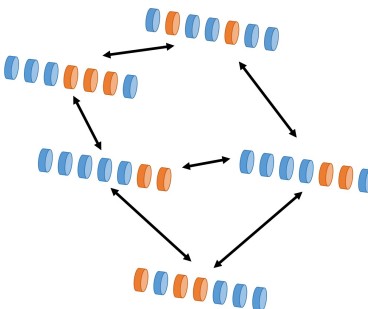 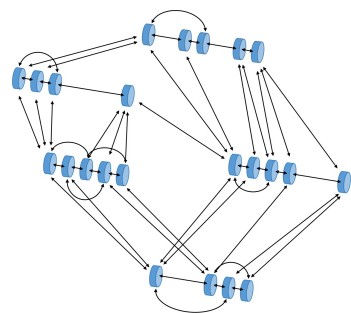

(a) Traditional graph structure for spatio-temporal forecasting with all data for a particular location considered as a single node. Orange features indicate missing values.

(b) Proposed graph formulation that does not separate space and time, but considers each recorded sample as an individual node. Missing values are simply omitted from the input data.

Figure 1: Visualisation of the traditional (a) and proposed (b) graph data structures for spatio-temporal forecasting. The graphs represent time series from five physical locations with, at most, $T = 7$ time-steps. The connectivity in both subplots was arbitrarily chosen and is simply meant for visualisation purposes.

## 4 Proposed Framework

### 4.1 New Unified Graph Formulation

Instead of treating each time series recorded at a particular location as a single node in a graph or each time-step as a separate graph, we propose to treat the measurements for a specific time and location as its own node. From this it follows that a graph, $\mathcal{G} = (\boldsymbol{X}, \boldsymbol{W})$, now have node features $\boldsymbol{X} \in \mathbb{R}^{M \times d_x}$, where $M \leq N \cdot T$, and edge weights $\boldsymbol{W} \in \mathbb{R}^{M \times M \times d_w}$. With no missing data, we have $T$ recorded samples for each of the $N$ locations, resulting in $M = N \cdot T$ nodes. However, in the presence of missing data or variable sequence lengths for different locations, our graph will have a smaller number of nodes, $M = \sum_i^N T_i$, where $T_i$ is the cardinality of the set of measurements at location $i$. Illustrations showing the traditional (described in Sec. 3.1) and proposed graph formulations are given in Fig. 1a and 1b, respectively. We here show some time series recorded for seven time-steps at five physical locations, with missing values shown in orange. It is clear how the proposed formulation in Fig. 1b represents spatial and temporal relations in the same manner, which means that spatial and temporal dependencies can be considered jointly. For the traditional setting in Fig. 1a, each node represents a physical location with some other adjacent locations, and missing values will have to be imputed or represented by a mask value. We see from Fig. 1a, that if one were to remove all series or time periods with missing information, there would be no input information left and a model would not be able to produce any meaningful forecasts. In our proposed graph structure shown in Fig. 1b, each recorded sample is considered as a node, which is connected to samples recorded either at the same or other physical locations. The connectivity shown in Fig. 1b could either follow some simple rules or be learned by the network, depending on the application. Nevertheless, an important feature of the proposed formulation is that missing values are simply not included nor required.

Now, consider the measurement locations in Fig 2, corresponding to 14 different meteorological measurement stations located in the North Sea. Assume that for every station, some meteorological variables are recorded every 10 minutes, such as wind speed, temperature and pressure. Each recorded value will be associated with a physical location and a timestamp. Since there is no temporal dimension in $\boldsymbol{X}$, we require temporal information to be directly encoded in the inputs, such as the encoding used for Transformers (Vaswani et al., 2017). This also applies to the physical location information. To encode the node features, we could have a suitable learned embedding for the timestamp, physical location and recorded values, by projecting them to a higher dimensional space, $d_x$. The latent representations could then be summed together to construct the

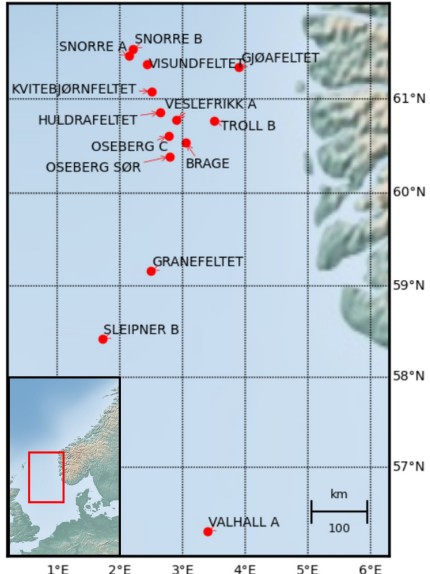

Figure 2: 14 meteorological measurement stations located in the North Sea.

particular node embedding as

$$
\begin{aligned}
\boldsymbol{X}_i = &\text{FeatEmbed}([ws_i \parallel p_i \parallel k_i])+ \\
&\text{PhysPosEmbed}([lat_i \parallel lon_i])+ \\
&\text{TimeEmbed}([ts_i]),
\end{aligned}
\tag{3}
$$

where $\parallel$ is the concatenation operator, FeatEmbed, PhysPosEmbed and TimeEmbed are some learned embeddings such as linear transforms to $d_x$ dimensional space. $ws_i$, $p_i$ and $k_i$ are the wind speed, pressure and temperature recorded at latitude $lat_i$, longitude $lon_i$ and timestamp $ts_i$, corresponding to a node, $i$. This was just a trivial example to show how recorded values, time and physical position information might be encoded into the node features, but various other embeddings could be used and should be adapted to suit the particular application. Similarly, if the problem also considers edge weight features, $\boldsymbol{W}$, the relative distance in both space and time between two nodes could be encoded as

$$
\boldsymbol{W}_{ij} = Embed([(lat_i - lat_j) \parallel (lon_i - lon_j) \parallel (ts_i - ts_j)]).
\tag{4}
$$

We also note that temporal and physical location information here is included in both eq. (3) and (4). However, only including such information in either the edge weights or node features might be sufficient. Since there are no constraints on varying sequence lengths, it follows that irregular time series or different sampling frequencies could be naturally included in a single graph. If there are measurements recorded with different frequencies, one could have distinct embeddings for different frequency data or include additional features to indicate particular sampling frequencies

## 4.2 Network Architecture

Models using the proposed graph formulation will be referred to as Spatio-Temporal Unified Graph Networks (STUGN). An illustration of the overall model architecture is shown in Fig. 3, where the graph block is the only necessary component and can be represented by any common type of GNN, such as the MPNN in eq. (2). In addition to graph updates, we also include the options for residual connections, normalisation and position-wise FFNs. In Fig. 3, $\mathcal{M}, \mathcal{P}$ and $\mathcal{T}$ are respectively the set of recorded values, physical locations and timestamps. To the embedding layer, we also feed $\mathcal{N}$, which defines which nodes should be connected to each other and is used to form the input edge weights $\boldsymbol{W}$. Inspired by Haugsdal et al. (2022), a persistence connection takes the last recorded value for a particular measurement location and adds it to the output,

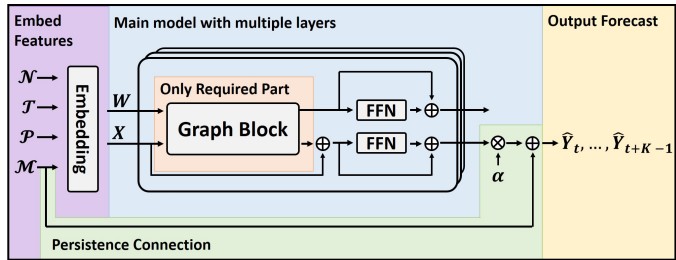

Figure 3: Generic spatio-temporal forecasting architecture. Depending on the graph structure and embedding, the illustration can represent both the traditional and proposed formulations discussed in Sec. 3 and 4, respectively. Only the embedding and graph blocks are strictly required.

which has been multiplied by some trainable parameter, $\alpha$, initialised to zero. This was found to speed up convergence and improve validation accuracy, as the models are initialised to a sensible starting point, namely the persistence model. Furthermore, instead of the layer normalisation commonly used in various Transformer architectures, we found that applying the ReZero (Bachlechner et al., 2021) initialisation after every Graph Block and FFN in Fig. 3 yielded better performance. In essence, the ReZero allows a network to learn the model depth by introducing gating into the residual connections.

Since the main motivation behind the study was to investigate the performance of the proposed unified graph formulation, instead of specific model architectures, we consider a couple of STUGN models using different graph update blocks. This was done to demonstrate that our framework is not limited to a single GNN architecture and to be able to better conclude on distinct characteristics of the proposed framework.

The first STUGN model will be denoted as the STUGN-GATv2, where the graph block in Fig. 3 is represented by the GATv2 (Brody et al., 2022), which is an improved version of the original GAT network (Veličković et al., 2017). For the second model, which will be referred to as the STUGN-TGAT, we propose an altered graph attention update based on the scaled dot-product attention (Vaswani et al., 2017) to attend to neighbouring nodes. Considering the graph update of the STUGN-TGAT model, a node, $i$, is updated based on information from neighbouring nodes $j \in \mathcal{N}_i$, through the following attention operation:

$$q_i = h_i^{(l-1)} \boldsymbol{W}^Q \tag{5}$$

$$k_{ji} = h_j^{(l-1)} \odot e_{ji}^{(l-1)} \boldsymbol{W}^K \tag{6}$$

$$\alpha_{ji} = \text{softmax}\left(\frac{k_{ji} q_i^T}{\sqrt{d_k}}\right) \tag{7}$$

$$h_i^{(l)} = \sum_{j \in \mathcal{N}_i} \alpha_{ji} h_j^{(l-1)} \boldsymbol{W}^V. \tag{8}$$

Here, $\boldsymbol{W}^{(\cdot)} \in \mathbb{R}^{d \times d_k}$ are trainable weight matrices, projecting from $d$ to $d_k$ dimensional space, and $\odot$ element-wise multiplication. Queries, $q_i$, contain the target node information while neighbouring nodes and corresponding edge information is embedded in the keys, $k_{ji}$. For input graphs without edge features, eq.(6) would be $k_{ji} = h_j^{(l-1)} \boldsymbol{W}^K$, and the key for a particular node would remain the same, irrespective of the target node, $i$. In eq. (6), edge features that describe the relationship between two nodes are transformed through $W^K$ and multiplied with the sender node features, $h_j^{(l-1)}$, to produce a more informative key representation. Instead of the linear transformation of edge features and element-wise multiplication with node features, one could concatenate the edge and node features before feeding these to a linear transform as in Zhang et al. (2022). However, the motivation behind our proposed key representation was to have a dynamic transform of neighbouring node features that depend on the edge weights. This would not be the case with an update like $k_{ji} = [h_j^{(l-1)} \parallel e_{ji}^{(l-1)}]\boldsymbol{W}^K$, where the key representation would be a sum of linear transforms of node and edge information separately. Even though detailed descriptions are not provided here, we experimented with different updates and found that the attention operation described above yielded good results. The

interested reader can check out our online repository at XXXX[1] for the complete implementation of a few different attention updates. Furthermore, the operations in eq. (5) - (8) are for a single attention head. In the actual update, we use multi-head attention, where heads are concatenated and transformed in exactly the same manner as in the vanilla Transformer implementation (Vaswani et al., 2017).

## 5 Experiments

### 5.1 Dataset

To demonstrate the forecasting performance of our proposed framework, we select the task of spatio-temporal wind speed forecasting due to its importance in facilitating the adoption of wind energy into power grids. Air temperature and pressure, wind speed and direction were recorded as 10-minute averages from June 2015 to February 2022, for the 14 locations in the North Sea shown in Fig. 2. Wind speed forecasts should be made for the next hour (6 steps ahead) for all locations simultaneously. The data was made available by the Norwegian Meteorological Institute and is openly available through the Frost API[2]. The first 60% of the data, with respect to time, was used for training, with the following 20% for validation and the last 20% for testing. Physical location information was represented by the latitude and longitude of a particular measurement station and all features were scaled to have zero mean and unit variance. Timestamp information for a recording was taken as the minute-of-hour, hour-of-day, day-of-month and month-of-year. For instance, values recorded at 18:50 on the 21st of February would have the timestamp $[50, 18, 21, 2]$. Each timestamp feature was further decomposed into its sine and cosine components. As an example, the hour-of-day stamp would be encoded as

$$[-1, 0] = [\sin(18 \cdot (2\pi/24)), \cos(18 \cdot (2\pi/24))]. \tag{9}$$

Some previous studies instead scale the timestamp information using a min-max scaler before feeding this to the embedding layer (Wu et al., 2021; Zhou et al., 2021). However, by not decomposing into sine and cosine components, the times 23:00 and 01:00 would be very far apart, despite the small two-hour difference. We, therefore, argue that our proposed method is better suited for encoding temporal information. Wind direction measurements were also decomposed into sine and cosine components to be able to represent the circular characteristic.

### 5.2 Baseline Methods for Comparison

The STUGN models were compared against four other baselines. Inspired by the important points raised by Zeng et al. (2022), the first two baselines were very simple models, namely the persistence and a one-layer linear model (TSF-Linear). The persistence model does not have any parameters and will simply extend the last recorded value at a particular station as the forecast for the next $K$ time-steps,

$$\hat{\boldsymbol{X}}_{:,t}, ... \hat{\boldsymbol{X}}_{:,t+K-1} = \boldsymbol{X}_{:,t-1}. \tag{10}$$

The TSF-Linear model, taken from Zeng et al. (2022), does not consider spatial dependencies and is simply a linear transform of the last $T$ recorded values for a station to the $K$ future predictions. Different to Zeng et al. (2022), we also found the best results when introducing a trainable persistence connection, as was done for the spatio-temporal models. The overall computation for the TSF-Linear model then becomes

$$\hat{\boldsymbol{X}}_{i,t}, ... \hat{\boldsymbol{X}}_{i,t+K-1} = \boldsymbol{Q} \boldsymbol{X}_{i,t-T:t-1} \cdot \alpha + \boldsymbol{X}_{i,t-1}, \tag{11}$$

where $\boldsymbol{Q} \in \mathbb{R}^{K \times T}$ is a linear layer along the temporal axis and $\alpha$ a trainable variable initialised to zero.

The other two baselines were more advanced, based on the traditional framework outlined in Sec. 3. The architectures were similar to that depicted in Fig 3, but with temporal update functions added subsequent to the graph updates. We will denote the two different baselines as ST-LSTM and ST-Transformer, depending on which temporal update function is used, LSTM or Transformer. With graph blocks represented by normal MPNNs (eq.(2)), the update and message functions were linear transforms with matrices $\phi \in \mathbb{R}^{2d \times d}$ and

---

[1] Will be made available in final version
[2] https://frost.met.no/index.html

Table 1: Final model specific parameters after tuning

| Parameter | STUGN | ST-LSTM | ST-Transformer |
|---|---|---|---|
| Learning Rate | 5e-05 | 1e-05 | 1e-05 |
| FFN Node | Yes | No | Yes |
| FFN Edge | [Yes, Yes, No]* | [Yes, Yes, No]* | [Yes, Yes, No]* |
| Normalisation | ReZero | - | Pre LayNorm |

\* Parameter values for the MPNN, GATv2 and TGAT settings, respectively.

$\psi \in \mathbb{R}^{3d \times d}$, where $d$ is the dimensionality of latent variables. Since each node, $i$, represents a physical location, eq. (2) only considers spatial relations. To also learn temporal correlations, the outputs from the MPNN-update are fed to either an LSTM or the full Self-Attention update from the Transformer, for the ST-LSTM and ST-Transformer, respectively. The operation of a Graph Block in Fig. 3 then becomes

$$h_{i,:}^{(l)} = g\left( \Big\|_{t=1}^{T} \left[ \phi\Big( h_{i,t}^{(l-1)} \| \bigoplus_{j \in \mathcal{N}_i} \psi(h_{i,t}^{(l-1)} \| h_{j,t}^{(l-1)} \| e_{jit}^{(l-1)}) \Big) \right] \right), \tag{12}$$

where $g$ is a temporal update function, $\bigoplus$ a mean aggregation of messages and $\|$ the concatenation operator. There are also options for the outputs, $h^{(l)}$ and $e^{(l)}$, to be added some residual connection and passed through FFNs before being outputted to the next layer, as shown in Fig. 3. More advanced graph updates were also tested, using the GATv2 and TGAT updates described in Sec. 4 instead of the MPNN update in eq. (12). For the ST-LSTM models, a direct strategy was employed, with forecasts for the next 6-steps ahead produced by projecting the final output through a two-layer MLP with 6 outputs. Considering the STUGN and ST-Transformer models, placeholders were used for the forecast locations in the inputs, set to the last available recorded values.

## 5.3 Experimental Set-up

To test the models under different amounts of missing values in the data, we construct four different sets of training, validation and test datasets, where we remove 0, 10, 20 and 30% of the entries from the original dataset. First, missing values are randomly sampled and then, if a value is missing, there is a probability, $p_n$, of the $n$ following values also being missing according to an exponential decay $p_n = \frac{exp(-n/10)}{\sum_{j=1}^{10} exp(-j/10)}$, with an upper limit of $n = 10$. However, since missing values are first sampled independently, before considering subsequent values, there still exist periods with more than ten subsequent missing values. For the baseline models, which require complete input data, missing values are interpolated to be in between the closest available values in the past and future. If no values are available in the inputs, missing values are set equal to those of the closest station, in geographical distance, with available data.

Since the STUGN framework naturally facilitates variable frequencies, we also construct an additional series for each location, which are the hourly mean wind speeds. Having both 10-minute and 1-hour frequency data could be an effective method for increasing the historical information used to make forecasts, without sacrificing detailed information in the short-term or significantly increasing the computational complexity. For the ST-LSTM and ST-Transformer models, the two series for a particular station were concatenated along the input feature dimension. If the two series had different lengths, the shortest would be padded with zeros to match the length of the longest series before concatenating. Look-back windows were the same for all models, set to the 18 and 12 previous time-steps for the 10-minute and 1-hour data, corresponding to 3 and 12 hours of historical information, respectively. All models used learned linear transforms to $d$ dimensional space as the embedding of temporal, physical position and measurement information, as in eq. (3). Since the ST-Transformer and STUGN models were without recurrence, sequence position information was also embedded into the node features, using the sine and cosine encoding from Vaswani et al. (2017). Input edge features contained physical distance information for all models, while the STUGN model also included temporal distance as in eq. (4). Hyper-parameter tuning was conducted based on the training and validation datasets with 30% missing values for all models. First, a wide search space was investigated using Bayesian search with Optuna (Akiba et al., 2019), before narrowing the search space and using grid-search

Table 2: Prediction accuracy in m/s on test dataset for 6-step (1-hour) ahead wind speed forecasts for measurement stations in Fig. 2.

| Percentage missing | 0 % | | 10 % | | 20 % | | 30 % | |
| Model | MSE | MAE | MSE | MAE | MSE | MAE | MSE | MAE |
|---|---|---|---|---|---|---|---|---|
| Persistence | 1.4478 | 0.8096 | 1.5029 | 0.8254 | 1.5660 | 0.8428 | 1.6262 | 0.8584 |
| TSF-Linear | 1.2843 | 0.7536 | 1.3278 | 0.7686 | 1.3739 | 0.7843 | 1.4215 | 0.7995 |
| ST-LSTM-MPNN | 0.9982 | 0.6775 | 1.0431 | 0.6943 | 1.0862 | 0.7107 | 1.1197 | 0.7234 |
| ST-LSTM-GATv2 | 1.0081 | 0.6808 | 1.0346 | 0.6914 | 1.0698 | 0.7047 | 1.1131 | 0.7212 |
| ST-LSTM-TGAT | 1.0433 | 0.6929 | 1.0769 | 0.7057 | 1.1192 | 0.7208 | 1.1639 | 0.7366 |
| ST-Transformer-MPNN | 1.0896 | 0.7057 | 1.1569 | 0.7291 | 1.1891 | 0.7438 | 1.2132 | 0.7509 |
| ST-Transformer-GATv2 | 1.0740 | 0.7002 | 1.0965 | 0.7091 | 1.1325 | 0.7238 | 1.1534 | 0.7335 |
| ST-Transformer-TGAT | 1.0996 | 0.7079 | 1.1506 | 0.7253 | 1.1706 | 0.7337 | 1.2155 | 0.7495 |
| STUGN-MPNN | 1.0647 | 0.6989 | 1.0962 | 0.7117 | 1.1350 | 0.7255 | 1.1656 | 0.7372 |
| STUGN-GATv2 | **0.9780** | **0.6678** | **1.0148** | **0.6827** | **1.0391** | **0.6923** | **1.0755** | **0.7058** |
| STUGN-TGAT | 0.9945 | 0.6743 | 1.0175 | 0.6847 | 1.0626 | 0.7009 | 1.1044 | 0.7165 |

to determine the final model parameters. All models had a latent dimensionality of 64 and consisted of three stacked layers (i.e. $l = 0, 1, 2, 3$ in eq. (2) and (12)). Model training was stopped after 25 epochs, using an Adam optimiser with a batch size of 16 and a 5% dropout rate on a single NVIDIA 2080Ti GPU. Considering Fig. 3, FFNs are assumed to be two-layer MLPs with a hidden layer dimensionality of 256 and GELU hidden activation. The outputs from the main models were passed through a final FFN without normalisation or residual connections before the outputs were added to a trainable persistence connection to produce the forecasts, as shown in Fig. 3. Any attention operation used four attention heads. The best values for the remaining model parameters were found to vary for the different architectures and are summarised in Table 1.

Graph connectivity, $\mathcal{N}$, was constructed based on the geographical distance between measurement locations so that every node was connected to its three closest measurement locations. For the STUGN models, the connectivity also spanned the temporal dimension and the neighbourhood for a particular node was set to the three closest nodes in terms of geographical distance and the three closest points along the temporal dimension in the future and past (i.e. $3 + 3 = 6$). Nodes that correspond to forecast locations, i.e. unknown values, did not send to any input nodes but received information from all other nodes corresponding to the same measurement location. The graph connectivity described here followed simple heuristics and we did not experiment with learnable adjacency matrices or more advanced metrics to determine node similarity and connectivity. This was left for future work, as the fundamental aim of this study was to demonstrate the effectiveness of the proposed framework under simple conditions, without engineering very domain-specific graph structures that might give the STUGN models an unfair advantage.

## 5.4 Results and Discussion

The mean squared error (MSE) was used to train the models, with the mean absolute error (MAE) also computed on the test data as

$$\text{MAE} = \frac{1}{n} \sum_{i=0}^{n} |y_i - \hat{y}_i| \tag{13}$$

$$\text{MSE} = \frac{1}{n} \sum_{i=0}^{n} (y_i - \hat{y}_i)^2, \tag{14}$$

where $y$ and $\hat{y}$ are the labels and predictions, respectively. Every model was trained for five different input seeds and with the four different missing data settings described previously. Predictive performance was evaluated based on the average from the five seeds, with the results given in Table 2, where the data was transformed back to a meters per second scale for better interpretability.

As expected, model errors increased as more values were removed. The TSF-Linear model was only a single linear transform of previous wind speeds to produce predictions for the next six steps ahead. In all

settings, the TSF-Linear model outperformed the Persistence model, meaning that it was able to learn some structures in the data to make forecasts, despite not being able to take spatial dependencies or complex non-linear relations into account. All other models significantly outperformed the TSF-Linear model, which proved their effectiveness in wind speed forecasting and indicated that the models were able to leverage spatial dependencies to improve forecast accuracies. Overall, using the GATv2 as the main graph update block yielded the best results for all models. However, considering the ST-LSTM models, the difference between the MPNN and GATv2 updates was very small and the MPNN update might therefore be preferred due to its simplicity. With higher test losses across all settings, the proposed TGAT update did not prove very effective for the LSTM-based models. For the ST-Transformer models, the TGAT updates achieved similar performance to those using the simpler MPNN updates, while the ST-Transformer-GATv2 achieved superior results across all data settings. Looking at the results in Table 2, it was found that the LSTM-based models outperformed all ST-Transformer models, indicating that the LSTM network might be more suitable as the temporal update function than a Transformer for short-term wind speed forecasting.

In contrast to the ST-Transformer and ST-LSTM models, the STUGN models did not have additional temporal networks, but with the MPNN, GATv2 or TGAT networks serving as the main update functions. It was found that the MPNN update, taking only a simple average aggregation of neighbouring features, was not very effective for the STUGN architecture. This was unsurprising since the graph operation for the STUGN models should be able to capture both complex temporal and spatial dependencies and the simple mean aggregation of the MPNN did not seem to be able to capture complex neighbourhood relations. On the other hand, when replacing the MPNN operation with TGAT or GATv2 networks, the STUGN architectures outperformed all other methods, with the STUGN-GATv2 model yielding the best results across all settings. Looking at the results for different amounts of missing data, the STUGN-GATv2 consistently achieved similar results to the ST-LSTM-GATv2, when the STUGN-GATv2 was subject to an additional 10 percentage points of missing data. For example, the MSEs and MAEs for the STUGN-GATv2 subject to 20% of missing data were similar to those for the ST-LSTM-GATv2 model under the 10% setting. Since both the STUGN-GATv2 and STUGN-TGAT models outperformed all other baselines, we argue for the effectiveness of the proposed unified graph formulation for spatio-temporal forecasting. Furthermore, even though the proposed graph formulation initially might seem more complex than the traditional framework used for the ST-LSTM and ST-Transformer models, the actual network architecture of the STUGN models is simpler than traditional methods as we alleviate the need for additional temporal networks. By removing the need for aligned inputs, the STUGN framework is also more versatile, being able to naturally facilitate irregular datasets or time series information for multiple sensors with potentially different or irregular sampling frequencies. This could be desirable for a range of forecasting systems where missing data might be a challenge or where sensors have different sampling frequencies.

To better understand the potential implications of the relative forecasting performances in Table 2, we estimate the associated energy production errors for the one-hour ahead forecasts. In Table 3, values are the absolute error improvements for estimated total energy produced, in kWh, compared to the Persistence model. Energy values were estimated by first transforming wind speed forecasts to kW using the power curve for the NREL 5 MW reference wind turbine (Jonkman et al., 2009). The 10-minute estimates for energy were then summed over the forecast interval to get the forecasted total energy production for the next hour. Results in Table 3 show the average improvements compared to the persistence model, where higher values are better. It was found that the STUGN-GATv2 models could improve accuracies by around 40 kWhs compared to the Persistence model and around 4-8 kWhs compared to the second-best model. Even though these were crude power estimates, it demonstrates some of the significant cost improvements that slightly more accurate forecasting models could achieve, especially as a wind farm would typically have a much larger capacity than 5 MW.

## 5.5 Future Work

Since this study was an initial investigation into a new unified spatio-temporal forecasting framework, there are a few areas that the authors note as particularly interesting for future work. Here, only wind speed forecasting was considered and the new graph formulation should be tested for different applications such as forecasting traffic, household energy consumption or sales data that are often irregularly sampled. The

Table 3: Estimated energy saving in kWhs for a single 5MW turbine per forecast horizon of 1 hour compared to the Persistence model.

| Percentage missing | 0 % | 10 % | 20 % | 30 % |
|---|---|---|---|---|
| Persistence | 0.00 | 0.00 | 0.00 | 0.00 |
| TSF-Linear | -2.27 | -2.26 | -2.20 | -2.97 |
| ST-LSTM-MPNN | 34.86 | 33.35 | 32.89 | 33.13 |
| ST-LSTM-GATv2 | 33.45 | 35.42 | 35.92 | 34.43 |
| ST-LSTM-TGAT | 28.01 | 28.38 | 28.16 | 27.12 |
| ST-Transformer-MPNN | 21.74 | 17.15 | 16.93 | 20.73 |
| ST-Transformer-GATv2 | 23.92 | 26.52 | 27.05 | 29.06 |
| ST-Transformer-TGAT | 20.02 | 18.71 | 22.32 | 21.09 |
| STUGN-MPNN | 25.22 | 25.52 | 26.53 | 27.34 |
| STUGN-GATv2 | **39.39** | **38.99** | **42.11** | **42.23** |
| STUGN-TGAT | 36.66 | 38.66 | 38.03 | 37.95 |

particular graph connectivity used might also be quite significant and studying methods for more advanced similarity evaluation of nodes or learnable graph adjacency matrices might be relevant to boost performance. Finally, self-supervised learning and pre-training have shown impressive results for a range of DL applications. Since the proposed graph formulation is very flexible and allows for a range of different frequency data or missing values, there is potentially a range of different self-supervised tasks that could be designed to improve downstream performance. Pre-training with a self-supervised task of predicting occluded values was experimented with for this paper, but did not yield significant accuracy improvements and therefore not included. However, the authors believe that more extensive research into particular self-supervised tasks might be useful, especially for applications where there might be more distinct trends and evident characteristics in the time series.

## 6 Conclusion

We propose a new graph formulation for spatio-temporal forecasting. By treating every recorded value as its own node in a graph, GNNs can be applied to learn both spatial and temporal dependencies jointly. The proposed STUGN models outperform more traditional methods for spatio-temporal forecasting that use GNNs in combination with temporal update functions. With the proposed graph structure, inputs do not have to be aligned along the temporal dimension and our framework naturally allows for missing values, irregular series and different sampling frequencies. Overall, this was intended as a study to investigate the feasibility of the proposed spatio-temporal formulation and we believe that future work would benefit from studying new applications, different from wind speed forecasting, as well as the effect of graph connectivity and feature embeddings.

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
