# OpenReview forum: "It is All Connected: A New Graph Formulation for Spatio-Temporal Forecasting"
_TMLR — Withdrawn by Authors_

### Review · Reviewer_Ab8i · 2023-05-30

**Summary Of Contributions:**

This paper studies spatio-temporal forecasting, a task that uses data from different locations and time periods, in various fields such as traffic, retail, energy, and weather. The focus is on short-term wind speed forecasting, using the North Sea as a case study due to planned offshore wind developments in the region.

The authors propose a new graph formulation using Graph Neural Networks (GNNs) to process spatial and temporal correlations simultaneously, an improvement over previous methods that used separate networks for these features. Their approach, the Spatio-Temporal Unified Graph Network (STUGN), offers several benefits:

- It reduces the need for additional temporal networks like Recurrent Neural Networks (RNNs) or Convolutional Neural Networks (CNNs).
- It tolerates missing or corrupt data, allowing for irregular time series and different sampling frequencies.
- It avoids shifting the data distribution by eliminating the need for imputation or sample removal when dealing with missing information.

The STUGN showed superior performance compared to other methods for spatio-temporal wind speed forecasting, even when dealing with incomplete training data. This demonstrates the effectiveness and potential of this new approach in handling real-world forecasting problems.

**Audience:**

Yes

**Claims And Evidence:**

Yes

**Requested Changes:**

1. In the experiment, this paper only introduces one dataset. Is it possible to have another dataset? This will make the results more convincing.
2. The experiment section mainly shows some quantitative results. It would be better if the authors could also present some qualitative results or case studies to intuitively demonstrate (1) why treating each recorded sample as an individual node is a better option, (2) why STUGN can well deal with missing data.
3. Although treating each recorded sample as an individual node is a neat idea, the size of the graph will become huge if there are a large number of recorded samples, resulting in much larger memory cost than traditional spatio-temporal forecasting architectures (e.g., ST-LSTM-MPNN). I would encourage the authors to discuss the issue of memory cost and how to apply STUGN to large-scale datasets with massive recorded samples.

**Strengths And Weaknesses:**

* Strengths
1. Innovative Approach: The paper introduces a novel Spatio-Temporal Unified Graph Network (STUGN) approach, which effectively incorporates both spatial and temporal features in a single Graph Neural Network. This innovative model simplifies the network structure and can potentially streamline computations.
2. Dealing with Missing Data: The STUGN can tolerate missing or corrupt data and does not require the input data to be aligned. This makes the approach more robust and practical for real-world applications where incomplete or imperfect data are common.
3. Proven Performance: The proposed model outperformed other baseline models in spatio-temporal wind speed forecasting tasks, demonstrating its practical effectiveness and potential for other similar applications.

Weaknesses:
1. Case-Specific: The research is focused on a specific case study, i.e., wind speed forecasting in the North Sea. Thus, the general applicability of the proposed STUGN to other forecasting problems or different geographical locations remains unproven.
2. Limited Contribution: Although the problem studied by the paper is interesting and the propose approach has some innovations, I think the overall contribution is limited in terms of the methodology. The major innovation is to consider each recorded sample as an individual node for modeling. This idea is not new and has been widely explored in natural language processing and data mining, where tokens in a sentence are treated as nodes, allowing one to apply a Graph Neural Network for modeling. Besides, the architecture of STUGN mainly follows existing GNN models, and there is not much new insight.
3. Insignificant Improvements: In the experiment, this paper compares STUGN against a variety of baseline methods. Although STUGN outperforms all the other methods, I feel like the improvement is not so significant.

---

### Review · Reviewer_K2RM · 2023-06-11

**Summary Of Contributions:**

This paper studies the problem of spatio-temporal time series forecasting and propose a method which views a record as a node. Then, a graph neural network is used to learn both temporal and spatial dependencies. Experiments on one dataset shows the compared results.

**Audience:**

No

**Claims And Evidence:**

No

**Requested Changes:**

1. Add more novel technical contributions and validate your argument.

2. Add more datasets.

3. Re-organize the related work part.

4. Add more baselines for comparison.

**Strengths And Weaknesses:**

Pros.

1. This paper studies an important problem.
2. The presentation is good.

Cons.
1. Viewing each observation as a node seems not to be novel, which can be seen in extensive existing methods [1]. Besides that, I didn't see any technical contribution.
2. The experiments are conducted on only one dataset, which makes the results less persuasive.
3. The related work part is not very clear, and I suggest to use "subsection".
4. More baselines are needed for performance comparison. (now is more like ablation studies)

[1] Coupled Graph ODE for Learning Interacting System Dynamics, KDD 2021

---

### Review · Reviewer_ekzz · 2023-06-13

**Summary Of Contributions:**

The paper addresses spatio-temporal forecasting, with application to wind speed prediction. Machine learning models applied to such data must consider both temporal and spatial information, while accommodating missing data, varying sampling frequencies, and irregular temporal patterns at different locations. The study explores deep learning architectures that treat each location as a node in a graph and employ graph neural networks to model the underlying structure. These architectures incorporate the temporal events recorded at each location as time series, using sequential models like LSTM or Transformers. However, the authors argue that such dual model architectures struggle with missing and irregular data, resulting in complex frameworks with increased training complexity. Instead of proposing a new architecture, the authors suggest an alternative data representation: each time point at a particular location is treated as a graph node. This eliminates sequential modeling, and each recorded sample becomes a node in the graph. A single graph neural network-based architecture can then jointly model the temporal and spatial information as nodes. The authors conduct experiments using wind speed data, reporting mean squared and absolute errors for short-term forecasting across various architectures. Baseline models include two linear models and different combinations of LSTM and transformer architectures in dual model spatio-temporal frameworks. Additionally, the authors qualitatively assess the implications of their approach by analyzing the associated energy production.

**Audience:**

Yes

**Broader Impact Concerns:**

The current exposition, in this limited form, does not present any immediate ethical concerns that need to be addressed as a part of this paper.

**Claims And Evidence:**

No

**Requested Changes:**

[Critical] It is important that authors provide a rigorous discussion and comparison with [1,2,3] and describe in detail how their work makes a significant technical contribution in light of. these works.

[Critical] The current results only provide a weak evidence of the authors claims about the utility of the new graph formulation for forecasting. It is recommended that authors present results for multiple domains and consider other metrics/qualitative analytics where they can showcase more significant impact of adopting this formulation.

[Minor] It is not clear why edge weights are assumed  to be fixed over time. While it make sense when considering wind mills and distance between them, it is a very restrictive assumption. Further, when they are fixed, why do authors choose to still use time-indexed edge wights? And finally what does edge weight signify for the edges between time points (represented as nodes)?

**Strengths And Weaknesses:**

### Strengths:
------------

- Spatio-temporal modeling is an important and a challenging topic and tit is great to see the work that applies its solution approaches to real-world systems.

- Missing and irregular data are certainly a big challenge in these settings and this paper’s focus on accounting for such information is very important.

- The paper does a good job in exploring various architecture choices pertaining to the basic deep learning blocks that account for both spatial and temporal information.


### Weaknesses:
---------------

- One of the major concerns I have with this paper is lack of discussion and comparison with significantly close works and it is difficult to discern the actual technical contribution of this work given the existing of these previous works [1, 2, 3]. The key contribution of this work is the new representation of spatio-temporal information so as to enable use of single GNN based architecture instead of separately modeling spatial and temporal information. The authors have cited [1] in their related work and mentioned that [1] already considers this idea but not in the application of forecasting. Additionally, [2] is a strong baseline for spatio-temporal forecasting with GNN. The authors do not cite or compare with this method which is a big miss. Finally, [3] is focussing on the same problem as addressed by the authors and have similar experiments and discussions as present in this paper. It is not clear how this work is different from [3] in terms of problem setting, solution approach and overall exposition.

- Given the above, it appears that this paper makes two contributions: investigates the application of an existing formulation of patio-temporal data to forecasting application and exploring the use of different existing deep learning architectures to  model such information. However, on both these fronts, the discussed results fall short of supporting the claims successfully. The results in Table 1 and Table 2 does not seem to be improving by statistically significant margin compared to some baselines (e.g ST LSTM in many cases) even for the highest level of missing data . Further, the authors just describe the results but does not provide any insights into why a particular architecture shows better results than other (E.g. why GAT performs best and where other variants fail? Why LSTM models are giving better results than Transformers in most dual model cases). As this paper mainly ends up studying the application of existing approaches to forecasting, it is important for the authors to provide detailed discussion and insights on these aspects

- Overall, I am not convinced by this approach (Atleast based on the current empirical evidence) of converting each time step into the node in the graph. I see two immediate problems: (i) scalability: This  could very easily become a very large graph and require high compute and big models to process the data effectively and may become prohibitive in real-world applications over time. For instance, sequential models can process temporal data in chunks while maintaining cross-chunk dependencies. Such an efficient solution is not trivial to achieve when all time points are considered as nodes in graph (ii) It is not clear why the nodes within a same time sequence end up learning almost similar representation to each other when they belong to the same location i.e. it is not clear how the authors ensure that nodes in time have different representations enough to distinguish temporal patterns.

- One benefit of using nodes in time sequence as graphs is stated as the ability to account for locations with missing or irregular data. The authors argue that in dual model case, an accepted approach is to completely discard such locations resulting in no predictive data available for them. But an interesting baseline in this case would be to pass the information from the neighboring locations (nodes in the graphs; GNN can do this trivially - think of imputing in the embedding space) to these locations and use that to predict the future. While this may not work for cases where entire sequence is missing, this has a lot of potential when some time points are missing but partial sequences are still available.

[1] Graph guided network for irregularly sampled time series. Zhang et. al. 2022

[2] SST-GNN: Simplified Spatio-temporal Traffic forecasting model using Graph Neural Network. Roy et. al. 2021

[3] Spatio-temporal wind speed forecasting using graph networks and novel Transformer architectures. Bentsen et. al. 2022

---

### Review · Reviewer_4JuF · 2023-06-15

**Summary Of Contributions:**

This paper studies spatio-temporal forecasting and propose a framework STUGN with an application to wind speed prediction. The proposed model is claimed to have the following benefits:
* learn spatial and temporal features jointly
* able to work with missing input information, irregular time series and different sampling frequencies.
* outperform many existing models for the task of spatio-temporal wind speed forecasting

**Audience:**

No

**Claims And Evidence:**

No

**Requested Changes:**

* explain how to apply this framework to other forecasting problems
* discuss the advantages of their strategy in detail and deal with the potential weak points of their model
* Minor: I think there is a typo in the definition of N_i on page 4

**Strengths And Weaknesses:**

### Strengths
* The proposed model learns spatial and temporal features jointly, while traditional models learn them separately.
* The proposed model inherits the advantage of GNN and does not require the features to be aligned, so it can naturally work with missing data, irregular time series, etc.

### Weaknesses
* [Evidence] The present findings indicate that STUGN performs effectively in predicting wind speeds. Nevertheless, the paper neither discusses how to apply this framework to other forecasting tasks nor presents STUGN surpasses other models in various forecasting problems. So the proposed framework's competitiveness is not convincingly established.
* [Technical] One of the main spirits of the proposed method is considering each recorded sample at a different timestep as a single node. However, it seems that this method will make the graph huge and fails to be applied to a big dataset. Moreover, it may require GNNs to possess a good ability to learn long-range dependency so that the proposed model can learn features through a large number of time steps.
* [Technical] In my opinion, utilizing an embedding to encode spatial and temporal features and learning the embedded features is more like a way to learn a "mixed" (sum) spatial and temporal feature rather than a joint one (concatenate). I cast doubt on the effectiveness of this method and question if learning spatial and temporal features jointly but updating them separately is an advantageous way. Nevertheless, this paper should persuade readers of the merits of their strategy.

---

### Note · Authors · 2023-06-25

**Comment:**

First of all, we would like to thank the dedicated reviewers for their time and very insightful feedback which will help improve the quality of our research. We agree with the reviewers that testing our STUGN framework on additional applications would help provide general confidence in the proposed methodology. The same argument also extends to comparisons against additional baselines.

A current drawback of the STUGN methodology, where each recording is represented as an individual node in a graph, is indeed that the graphs might become very large and result in significant computational and memory cost. We therefore agree that some further discussion of potential alterations to the architecture, which might help avoid this overhead, could be useful. For instance, the authors believe that a hierarchical approach could be particularly interesting. In such a setting, input neighbourhood information (in space and time) could first be aggregated to a single node representation in a lower-resolution graph. Latent features from processing on such a lower-resolution graph could then be used as inputs to higher-resolution subgraphs to produce the final high-resolution forecasts.

At this time, we have decided to retract the paper due to the limited experiments on the versatility of the proposed framework for other applications. This would take some time to implement and we therefore believe it better to retract the paper to conduct further research, before the paper could be potentially suitable for publication in an ML journal like TMLR.

Thanks again to all the reviewers and action editor for their time in reviewing the paper and providing invaluable insights which will inevitably help improve the research.


**Withdrawal Confirmation:**

I have read and agree with the venue's withdrawal policy on behalf of myself and my co-authors.